# Administration of Estradiol Benzoate Enhances Ovarian and Uterine Hemodynamics in Postpartum Dairy Buffaloes

**DOI:** 10.3390/ani13142340

**Published:** 2023-07-18

**Authors:** Haney Samir, Hossam R. El-Sherbiny, Ahmed Ezzat Ahmed, Ramya Ahmad Sindi, Khalid M. Al Syaad, Elshymaa A. Abdelnaby

**Affiliations:** 1Theriogenology Department, Faculty of Veterinary Medicine, Cairo University, Giza 12211, Egypt; hossamelsherbiny7353575@cu.edu.eg (H.R.E.-S.); elshymaa.ahmed@cu.edu.eg (E.A.A.); 2Biology Department, College of Science, King Khalid University, Abha 61413, Saudi Arabia; aabdelrahman@kku.edu.sa (A.E.A.); alsyaad@kku.edu.sa (K.M.A.S.); 3Department of Laboratory Medicine, Faculty of Applied Medical Sciences, Umm Al-Qura University, Makkah Al-Mukarramah 24382, Saudi Arabia; rahsindi@uqu.edu.sa; 4Department of Clinical Sciences, College of Veterinary Medicine, King Faisal University, Alahsa 31982, Saudi Arabia

**Keywords:** buffaloes, estradiol benzoate, color Doppler ultrasonography, nitric oxide, ovarian and uterine hemodynamics, postpartum

## Abstract

**Simple Summary:**

Buffaloes (*Bubalus bubalis*) are one of the important dairy animals in Egypt because they account for about 56% of total milk production. However, Egyptian buffaloes are characterized by low reproductive potential. Prolonged acyclicity and anestrum are major sources of economic loss in postpartum buffaloes. Therefore, various protocols that have been established to enhance the reproductive performance of postpartum buffaloes are of great value. The purpose of the present study was to assess the effect of the administration of estradiol benzoate on ovarian and uterine hemodynamics using color Doppler technology in postpartum dairy buffaloes. The findings of the current study are very important because they give new insight into the potential improvement of estradiol benzoate for improving the hemodynamic parameters of ovarian structures and the uterus concomitantly, with improvements in circulating estradiol and nitric oxide levels, in buffaloes 4 weeks postpartum.

**Abstract:**

The postpartum (PP) period is a crucial stage for the resumption of reproductive performance and ovarian cyclicity in dairy buffaloes. The present study aimed, for the first time, to assess the effect of the administration of estradiol benzoate (EB) on ovarian and uterine hemodynamics in PP dairy buffaloes. Eight pluriparous acyclic domestic buffaloes were enrolled in the present experiment and received a dose of 10 mg of estradiol benzoate (EB) intramuscularly 4 weeks after parturition. All animals were examined two times before EB administration (days −3, and −1) and on the day of EB administration (day 0), followed by examinations on days 1, 3, 5, 7, and 9 post-EB administration. The middle uterine artery (MUA) and ovarian artery (OA) blood flow patterns were assessed using a color Doppler ultrasound device. The reproductive parameters were (1) the cross-sectional diameters (cm) of the OA and MUA, (2) cranial uterine horn thickness (UHT; cm), and (3) hemodynamic changes within the MUA on both the ipsi- and contra-lateral sides of the previous pregnant horn and within the OA corresponding to the ovarian tissues. The examined blood flow parameters were the pulsatility index (PI), resistance index (RI), peak systolic/end-diastolic ratio (S/D), time-averaged maximum velocity (TAV; cm/s), uterine blood flow rate (BFR; bpm), and uterine blood flow volume (BFV; mL/min). Concomitantly, blood samples were collected from the coccygeal vein, and the sera were stored at −18 °C for use in estradiol (E2-17β) and nitric oxide (NO) assays. The results revealed increases in both OA and MUA cross-sectional diameter (cm) on the ipsi-lateral and contra-lateral (*p* < 0.05) sides within 24 h until day 9 post-treatment. The values of the RI and PI of blood flow within the OA and MUA on the ipsi-lateral and contra-lateral sides of the previous pregnancy were obviously lower (*p* < 0.05) at 24 h after the administration of EB, and then, started to gradually elevate, reaching the pre-treatment values on day 9 after EB administration. Both the BFR and BFV in the OA and MUA significantly increased from 24 h to 72 h after EB administration on both the ipsi-lateral and contra-lateral sides (*p* < 0.05); then, their values started to decrease to reach the pretreatment value on day 9 after EB administration. Both E2 and NO concentrations significantly increased (*p* < 0.05) from 24 h until day 3 after EB injection and then started to decline after that, reaching the pre-treatment value on day 9. In conclusion, the administration of EB enhances the ovarian and uterine blood flow concomitantly with increased levels of NO in PP dairy buffaloes.

## 1. Introduction

Buffaloes (*Bubalus bubalis*) are one of the important dairy animals in Egypt (with a population of 3.69 million) because they account for about 56% of total milk production [1]. Compared to cows, Egyptian buffaloes show greater adaptability to the harsh sub-tropical environment and resistance to various diseases [2,3]. However, buffaloes raised in Egypt are characterized by low reproductive potential (seasonality, faint estrus symptoms, varied ovulation times, inter-estrous intervals, silent estrus, and long inter-calving intervals) relative to that of cows [4]. Prolonged acyclicity (up to 90 days) and the anestrus period are major sources of economic loss in postpartum buffaloes [4,5]. Under favorable conditions, the involution of the uterus in buffaloes is usually completed in 25–35 days after parturition [6]. Due to the involution process, uterine blood flow decreases linearly during the postpartum period [7]. Indeed, the blood flow of female reproductive organs, especially the uterus and ovaries, plays a crucial role in reproductive potential in cattle [8]. A close relationship between ovarian blood flow and the successful establishment of pregnancy was verified in cows [9] and buffaloes [10]. Therefore, various protocols that have been established to enhance blood flow to the reproductive tract are very important for the resumption of the reproductive performance of postpartum buffaloes. Taken together, our aim is to administer an agent that can enhance uterine and ovarian blood flow in postpartum buffaloes after the completion of uterine involution (around 4 weeks postpartum). Physiologically, estradiol acts on the hypothalamus to induce behavioral estrus and stimulate a surge in luteinizing hormone (LH) to induce ovulation [11]. Several pieces of evidence in cattle have revealed the pivotal roles of preovulatory estradiol in the growth of follicles, oocyte maturation, sperm transport, and finally, the embryo survival rate [12,13]. In addition, several studies have reported the vasodilatory effect of estradiol and its pivotal role in reproductive organs’ blood perfusion [14,15]. However, the usefulness of estradiol in ovarian and uterine hemodynamics has not been studied in postpartum buffaloes after the completion of uterine involution.

Since the 1980s, ultrasonographic evaluations have been incorporated into various veterinary practices, such as the determination of pregnancy status [16], the assessment of normal embryo/fetal development [17], and the monitoring of postpartum uterine involution [18]. The use of the color Doppler ultrasonography technique has enhanced the diagnostic reliability of ultrasonography in farm animal reproductive practices because it enables the assessment of morphological and functional aspects of the organ, based on its blood perfusion [9,19]. In buffaloes, the Doppler ultrasonographic assessment of ovarian and uterine hemodynamics has been frequently used to identify the early pregnancy status based on either the objective [20,21] or subjective [10] assessment of the corpus luteum and/or uterine blood flow. Additionally, Doppler sonographic examination has been used for monitoring the postpartum involution of the uterus in cows [22] and small ruminants [23]. However, its usefulness in assessing ovarian and uterine hemodynamics after the administration of EB has not been investigated in postpartum buffaloes. With these considerations, the present study aimed to assess the effect of the administration of estradiol benzoate (EB) on ovarian and uterine hemodynamics in postpartum dairy buffaloes. In addition, the circulating levels of estradiol and nitric oxide were evaluated in the sera because of their pivotal roles in ovarian and uterine hemodynamics [24,25].

## 2. Materials and Methods

### 2.1. Animals

Buffaloes in this study were managed, treated, and used for research following ethical approval from the Animal Care and Use Committee belonging to Faculty of Veterinary Medicine, Cairo University (Vet CU-03162023670). All methods were performed following ARRIVE guidelines.

### 2.2. Experimental Design, Housing, and Management

The current work was conducted at the large animal research farm, Theriogenology Dep., Faculty of Veterinary Medicine, Cairo University (30.0276° N, 31.2101° E). Eight healthy puerperal and pluriparous domestic buffaloes (*Bubalus bubalis*; 500 ± 50 kg BW and 5–7 years of age) were enrolled in the present experiment. Parturition’s history revealed that there were no interventions to assist the delivery process of all animals. Buffaloes were kept indoors, fed on a commercial pelleted ration according to NRC guidelines, and accessed fresh water freely. All females had been acyclic (buffaloes until then showed no signs of resumption of ovarian cyclic activity [4,5]) at the start of the experiment and were observed each other day starting from the 4th week (30 ± 2 days) postpartum. All buffaloes had no abnormal vaginal discharges or uterine or udder pathologies [26]. All buffaloes were administered a single intramuscular dose of 10 mg of estradiol benzoate (2 mg of estradiol benzoate in every 1 mL, Ovahormon, Asuka Animal Health Co., Ltd., Tokyo, Japan). Animals were examined two times before EB administration (days −3 and −1) and on the day of EB administration (day 0), followed by examinations on days 1, 3, 5, 7, and 9 post-EB administration [27].

### 2.3. Ultrasonographic Examination (B-Mode, Color, and Pulsed-Wave Doppler)

Identification of a previous pregnant uterine horn was based on the farm records during pregnancy diagnosis using rectal palpation to differentiate the ipsi-lateral uterine artery from the contra-lateral one. The middle uterine artery (MUA) and ovarian artery (OA) blood flow patterns were assessed using a color Doppler ultrasound device (EXAGO, France, equipped with a 6–12 MHz linear transducer). Reproductive tract assessment included the following:(1)Cross-sectional diameters (cm) of the ovarian artery (OA; Figure 1A,B) and MUA (Figure 1C,D) were measured using the electronic caliber of an ultrasound device after observing each artery via B-mode ultrasonography.(2)Cranial uterine horn thickness (UHT; cm) was measured as previously proposed [28]. In brief, after observing the uterine horn via B-mode ultrasonography, the transverse diameter of the anterior third section of each horn (proximal to the uterine body) was measured via the electronic caliber of the ultrasound device.(3)Hemodynamic changes within the MUA on the ipsi- and contra-lateral sides of the uterine horn at its origin (the internal iliac artery) and within the OA corresponding to the ovarian tissues were monitored as previously reported [29,30] (Figure 2). In brief, once the MUA was visualized via B-mode ultrasonography, blood flow was affirmed using the Doppler device color mode, followed by pulsed-wave mode activation for Doppler index measurement. All calculations were determined automatically by the device and measured three times to obtain the mean by the same person. At least three successive waves (cardiac cycles) were estimated to measure the Doppler parameters with calibrated and fixed Doppler settings throughout the study, as follows: The pulse repetition frequency was 3.5 KHz, the angle of insonation was less than 60 °, and the color flow mapping included two colors (red and blue; Figure 2). Examined blood flow parameters were the pulsatility index (PI), resistance index (RI), peak systolic/end-diastolic ratio (S/D), time-averaged maximum velocity (TAV; cm/s), uterine blood flow rate (BFR; bpm), and uterine blood flow volume (BFV; mL/min). BFV was estimated using the following equation: BFV = TAV(cm/s) × (D(cm) × 0.5)^2^ × π 60 [22].

### 2.4. Blood Sampling and Assessment of E2 and NO Concentrations

The collection of blood samples was performed via the coccygeal vein using sterile vacutainers, followed by centrifugation at 1800× *g* for 10 min, and the harvested sera were stored at −18 °C for the assessment of E2 and NO concentrations. An E2 (E2-17β) assay was performed using a competitive type of ELISA (DRG kits, Marburg, Germany), with a sensitivity of 9.8 pg/mL [31]. Nitric oxide (NO) was determined colorimetrically using a nitric oxide kit (Bio-diagnostics, Giza, Egypt) with a sensitivity of 0.24 μmol/L [32,33].

### 2.5. Statistical Analysis

At first, all data were normally distributed according to the Shapiro–Wilk test. For each side of the MUA and OA (ipsi-lateral or contra-lateral), blood flow parameters (RI, PI, S/D, BFV, and BFR), diameters of the MUA and OV, and the UHT of all buffaloes were pooled/time point. A repeated measures ANOVA was assigned to compare the means of the examined parameters from day −3 until day 9 after estradiol benzoate injection, followed by Bonferroni’s post hoc analysis. Results were expressed as means ± SEM. Pearson correlation was performed between Doppler parameters (RI, PI, S/D, BFR, and BFV), MUA diameter, UH thickness, and E2 levels (r). The significant difference was set at *p* < 0.05. Data were analyzed using SPSS (analysis software program, version 16, Chicago, IL, USA).

## 3. Results

### 3.1. Alterations in the Ovarian and Uterine Arterial Diameters, and Uterine Horn Thickness

Both the OA and MUA cross-sectional diameter (cm) increased (*p* < 0.05) on the ipsi-lateral and contra-lateral sides within 24 h until day 9 post-treatment (Figure 3A,B), while uterine horn thickness (UHT; cm) was not affected by the EB injection over the studied time points (Figure 3C). In addition, both sides’ (ipsi-lateral and contra-lateral) values of cross-sectional diameter and UHT were positively correlated with each other (r = 0.72, *p* < 0.05).

### 3.2. Alterations in the OA and MUA PI, RI, and S/D Ratio

All Doppler indices (PI, RI, and S/D) had significantly (*p* < 0.05) lower values 24 h after EB administration on both sides (ipsi-lateral and contra-lateral sides) of the OA and MUA, and then started to elevate reaching the pre-treatment values on day 9 post-EB injection (Figure 4). All calculated Doppler indices of the blood flow (RI, PI, and S/D) within the OA and MUA showed non-significant differences between the ipsi-lateral and contra-lateral sides (*p* > 0.05). Significant positive correlations were detected between both sides in terms of PI values (r = 0.97 and *p* < 0.01 for OA; r = 0.93 and *p* < 0.01 for MUA) and RI values (r = 0.852 and *p* < 0.01 for OA; r = 0.77 and *p* < 0.01 for MUA), and S/D ratios (r = 0.85 and *p* < 0.01 for OA; r = 0.87 and *p* < 0.01 for MUA).

### 3.3. Alterations in the OA and MUA BFR (bpm), and BFV (mL/min)

BFR and BFV within the OA and MUA increased on both sides on day 1 until day 3 post-injection of EB (*p* < 0.05), while their values started to decrease after day 3 and reached the pretreatment value at day 9 post-injection (Figure 5B). No significant differences in the BFR and BFV between the ipsi-lateral and contra-lateral sides of the two assessed arteries were found. Values of BFR and BFV in the two assessed arteries (OA and MUA) showed strong positive correlations between the ipsi-lateral and contra-lateral sides (r = 0.89 and *p* < 0.01, and r = 0.99 and *p* < 0.01 for OA; r = 0.84 and *p* < 0.01, and r = 0.86 and *p* < 0.01 for MUA).

### 3.4. Alterations in the E2 and NO Concentrations

Both E2 and NO concentrations significantly increased (*p* < 0.05) within 24 h after estradiol benzoate injection, continued to be elevated until day 3 and, then started to decline after that, reaching the pre-treatment value on day 9 post-EB administration (Figure 6). High positive correlations were detected between E2 and NO concentrations (r = 0.85 and *p* < 0.01), and between E2 concentrations and BFR of the OA (r = 0.94 and *p* < 0.01) and the MUA (r = 0.77 and *p* < 0.01). However, high negative correlations were found between the concentrations of E2 and the Doppler indices of blood flow within the OA (r = −0.84 and *p* < 0.01 for PI; r = −0.81 and *p* < 0.01 for RI; r = −0.91 and *p* < 0.01 for S/D) and the MUA (r = −0.67 and *p* < 0.05 for PI; r = −0.68 and *p* < 0.05 for RI; r = −0.79 and *p* < 0.01 for S/D).

## 4. Discussion

The period of postpartum anoestrus or acyclicity in buffaloes is usually longer than that in cattle [5]. Indeed, various regimens are intended to synchronize estrus and activate ovarian cyclicity in postpartum buffaloes, such as the OvSynch protocol. In this regimen, a first dose of gonadotropin-releasing hormone (GnRH) is followed by a dose of prostaglandin F2 alpha (PGF_2α_), and finally, a second shot of GnRH is administered [12] to synchronize estrus. However, this regimen has some limitations, such as the ovulation of sub-mature follicles, that may lead to a reduction in pregnancy rates [13]. Until now, and according to the best of the authors’ knowledge, the present study is the first to assess the impact of EB administration on uterine and ovarian hemodynamics in four-week-post-partum buffaloes (after the completion of uterine involution). Several reports have revealed the potential impacts of EB on cattle reproduction. In the absence of the corpus luteum, the administration of estradiol could elevate preovulatory estradiol concentrations, which are important for follicular development, ovulation success, and the establishment of pregnancy [34,35,36]. Ovariectomized cows treated with estradiol before embryo transfer were more likely to maintain a pregnancy until day 29 than ovariectomized cows not treated with estradiol [37]. The results of the current study supported the hypothesis that EB enhanced the ovarian and uterine hemodynamic parameters as assessed via color Doppler ultrasonography. In the present study, we assessed the Doppler indices (RI, PI, and S/D). These indices do not represent the direct measurement of organ blood flow but rather describe blood flow resistance in the examined vessels. These indices are angle-independent and indicate the flow condition downstream [38,39]. Negative correlations were reported between Doppler indices (RI and PI) and the vascular perfusion of tissue downstream [40,41]. As the values of Doppler indices increase, the resistance to blood flow increases, and vice versa [19,42]. Therefore, decreased values of Doppler indices indicated increases in blood perfusion. These changes in Doppler index values are important if there is a need for continuous oxygen and nutrient supplies to the respective organ [43,44]. In the present study, EB induced significant decreases in Doppler indices of the OA and MUA. Decreased Doppler indices indicate decreases in blood flow resistance and refer to an increase in blood perfusion and a continuous supply of oxygen and nutrients to the respective organs [19,42].

The mechanism through which EB influences ovarian and uterine hemodynamics (as represented by increases in both BFR and BFV and decreases in the measured Doppler indices of the OA and MUA blood flow) was not elucidated in the current study with certainty. However, two possible pathways may be considered to interpret these findings. Firstly, EB could modulate uterine and ovarian hemodynamics through its direct effect since there were increases in the concentrations of E2, which were concomitant with decreases in the values of RI and PI of the measured arteries. Estradiol has been reported to have a strong vasodilatory effect on the blood perfusion of reproductive organs [15,45,46]. In cows, the administration of EB on the day before the expected estrus increased the blood perfusion of the endometrium but did not result in significant changes in the characteristics of ovarian structures [47]. In the present study, E2 concentrations were negatively associated with the values of both the RI and PI of the OA and MUA. Similarly, strong correlations between Doppler indices of testicular hemodynamics and E2 concentrations were reported in goats [47,48], rams [32,33], buffalo bulls [3], and stallions [45]. Results of the present study indicated an associative relationship between E2 and Doppler indices, an observation deemed important, but there was no elucidation of the underlying mechanism for E2 action. Additionally, in women, uterine and ovarian blood flow is regulated by sex steroids, and the exogenous administration of these hormones resulted in increases in the blood perfusion of the uterus and ovary [42]. The vasodilatory effect of E2 may be mediated through intracellular signaling, which involves decreases in the calcium uptake of potential sensory channels caused by the E2 receptors in the tunica media and E2 in the uterine arteries [49].

In the current study, strong correlations were found between NO levels and the studied Doppler parameters of blood flow within the OA and MUA. Therefore, another explanation for increased OA and MUA blood flow in this study could be attributable to increased levels of NO, which is considered one of the most powerful vasodilators [50,51]. Nitric oxide is synthesized from L-arginine methyl ester amino acid by the NO synthase enzyme in various peripheral tissues, including the ovarian and uterine vasculature and seminiferous tubules [24]. Some studies have proposed the local effects of NO on regulating the distribution of oxygen, nutrients, and hormones via the ovarian and uterine vessels [24,25,52,53]. Nitric oxide (NO) regulates vascular tone and blood flow by activating soluble guanylate cyclase (sGC) in vascular smooth muscle and controls the consumption of oxygen inside mitochondria by inhibiting the cytochrome c oxidase enzyme [53]. In rat models, locally produced NO plays a great role in the maintenance and increase of ovarian blood flow during the preovulatory period [25]. Nitric oxide deficiency could inhibit the release of GnRH and disturb FSH and LH release. A chronic deficiency of NO minimized the release of GnRH in the hypothalamus of female rats, accompanied by decreases in LH and FSH secretions [54]. The NO intracellular pathway could be mediated through its binding to soluble guanylate cyclase (sGC) and activating sGC to increase cellular cyclic guanosine monophosphate (cGMP) concentrations [24]. When the concentration of cGMP increases, the activation of cGMP-dependent protein kinase G induces a decrease in intracellular calcium concentration, which mediates vascular smooth muscle relaxation, platelet aggregation inhibition, vascular permeability increase, and other biological effects [52]. In the female reproductive system, NO affects reproductive processes such as follicular development, oocyte maturation, ovulation, luteinization, fertilization, embryo development, pregnancy maintenance, childbirth, and the regulation of the menstrual cycle. At the same time, lacking NO secretion during pregnancy can lead to fetal abnormalities [24]. However, how exactly the mechanism of EB administration could modulate OA and MUA blood flow was not elucidated in the present study, and further research may be needed.

This study was performed on four-week-postpartum buffaloes. During this stage and up to 90 days postpartum, animals were acyclic [4,5] and had low blood flow to the ovary and the uterus because of the completion of uterine involution [7,55]. Therefore, our target was to assess whether the administration of EB could alter the hemodynamic parameters of the ovary and the uterus or not. Furthermore, the authors thought that comparing the studied parameters before and after the administration of EB would be appropriate to test the effects of EB on the same individual buffaloes to avoid individual variations. However, the inclusion of a control group and extending the plan to cover fertility potential may be important in further studies on a large-scale population of buffaloes.

## 5. Conclusions

In conclusion, the administration of EB to 4-week-postpartum acyclic dairy buffaloes increased the cross-sectional diameters of the ovarian and middle uterine arteries. Concomitantly, most of the studied parameters of ovarian and uterine blood flow were enhanced. In addition, EB increased levels of E2 and NO in the postpartum dairy buffaloes. The findings of this study present important knowledge because they give new insight into the potential improvement of EB in the detailed hemodynamic parameters of the ovary and the uterus in 4-week-postpartum buffaloes. However, the application of this protocol to a large population, considering the possibility of assessing fertility potentials (for example, pregnancy rates), may be interesting in a further study.

## Figures and Tables

**Figure 1 animals-13-02340-f001:**
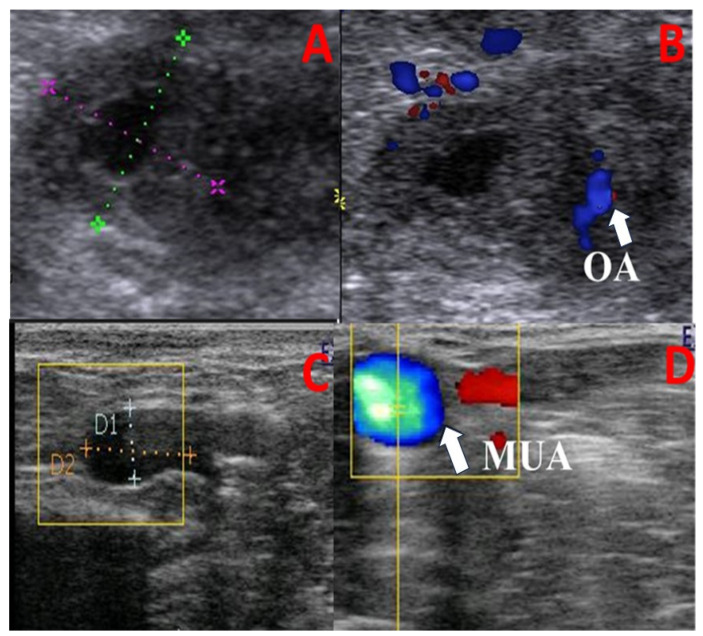
Ultrasonograms showing the ovarian tissue in the B-mode (**A**), ovarian artery vascularization in the color mode (**B**) illustrated by the white arrow, middle uterine artery cross-sectional diameter in the B-mode ((**C**); MUA; cm), and MUA vascularization in the color mode (**D**) illustrated by the white arrow. OA = ovarian artery, and MUA = middle uterine artery. Different-colored dots in Subfigure (**A**) refer to the ovarian boundaries, while the different-colored dots in the square in Subfigure (**C**) refer to the measurement of ovarian artery diameter.

**Figure 2 animals-13-02340-f002:**
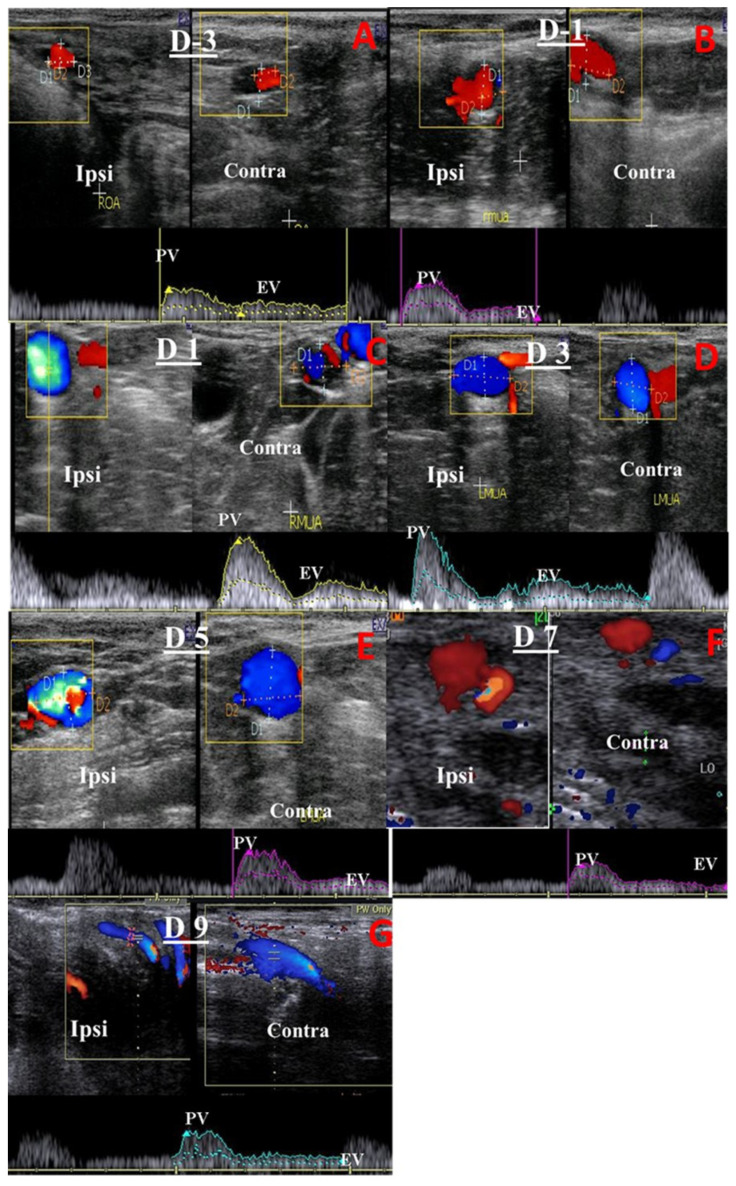
Hemodynamic changes on the ipsi-lateral and contra-lateral sides of the middle uterine artery (MUA) were monitored using a color pulsed-wave Doppler device on days −3 (**A**), and –1 (**B**), and 1 (**C**), 3 (**D**), 5 (**E**), 7 (**F**), and 9 (**G**) post-estradiol benzoate administration in puerperal buffaloes. The squared content in subfigures represents the imaging of the MUA via color Doppler ultrasonography. ipsi = ipsi-lateral, Contra = contra-lateral, PV = peak velocity point, and EV = end velocity point.

**Figure 3 animals-13-02340-f003:**
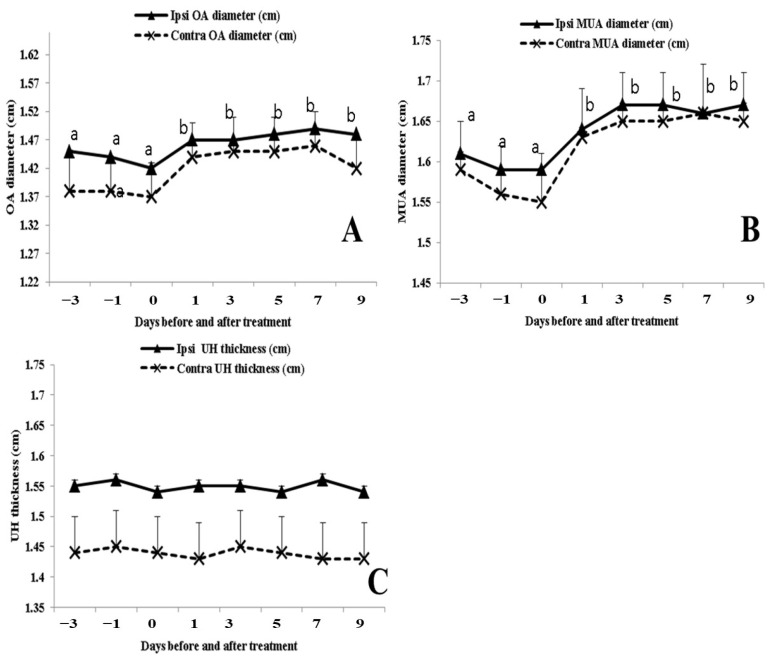
Changes in ovarian artery diameter (OA. diameter; cm; (**A**)), middle uterine artery cross-sectional diameter (MUA; cm; (**B**)), and uterine horn thickness (UHT; cm; (**C**)) in buffalos (*n* = 8) from 3 days before until 9 days after estradiol benzoate injection. Values are presented as mean ± SEM and different superscript values indicate the existence of significant differences (*p* < 0.05).

**Figure 4 animals-13-02340-f004:**
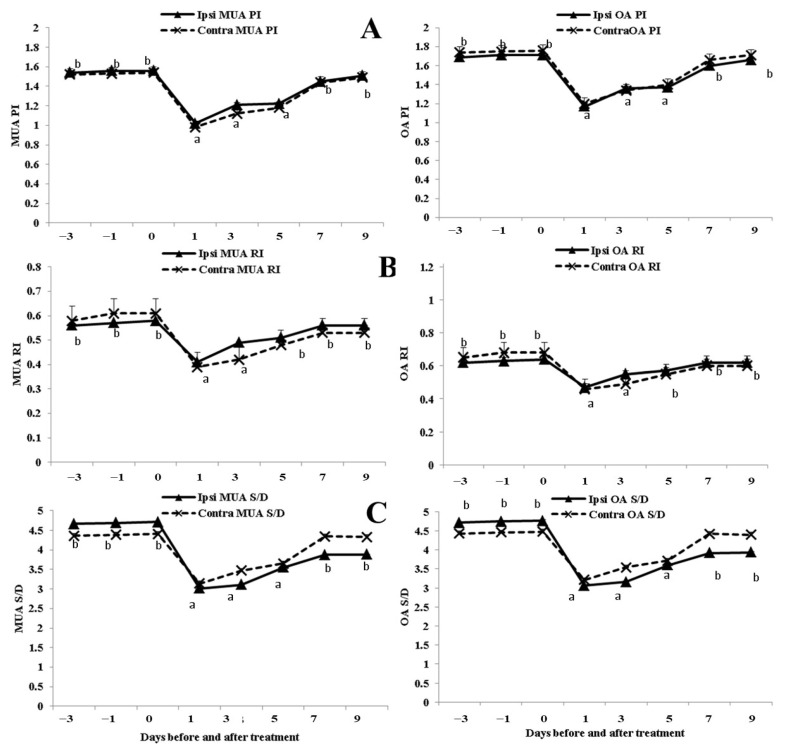
Changes in ovarian and uterine pulsatility index (PI; (**A**)), resistance index (RI; (**B**), and systolic/diastolic ratio (S/D; (**C**)) in buffaloes (*n* = 8) from day −3 until day 9 after estradiol benzoate injection. Values are presented as mean ± SEM. Different superscript values indicate significant differences (*p* < 0.05).

**Figure 5 animals-13-02340-f005:**
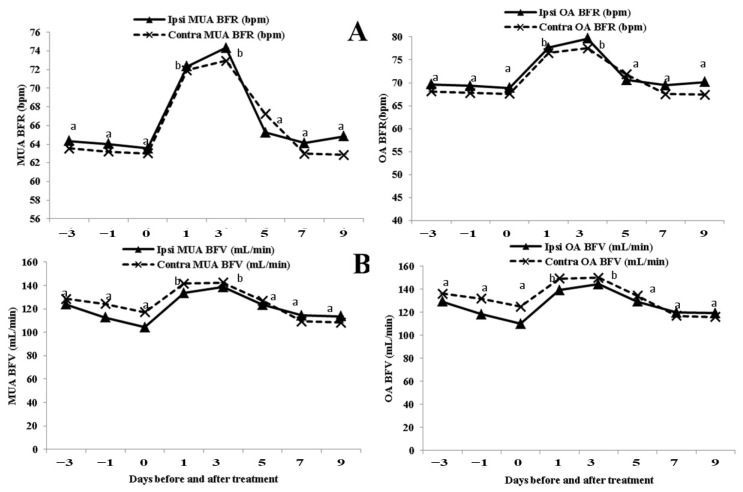
Changes in ovarian and uterine blood flow rate (BFR; bpm; (**A**)), and blood flow volume (BFV; mL/min; (**B**)) in buffaloes (*n* = 8) 3 days before until 9 days after estradiol benzoate injection. Values are presented as mean ± SEM and different superscript values indicate significant differences (*p* < 0.05).

**Figure 6 animals-13-02340-f006:**
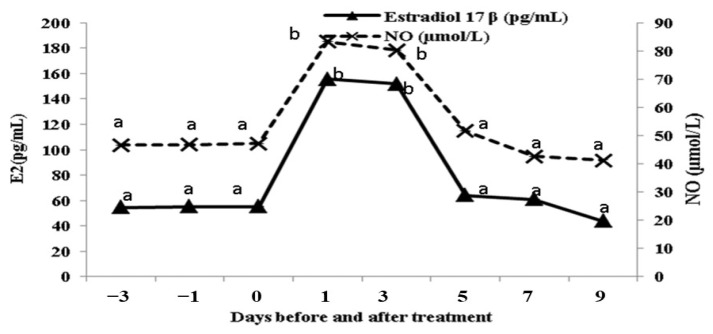
Changes in estradiol 17ß (E2; pg/mL), and nitric oxide (NO; µmol/L) concentrations in buffalos (*n* = 8) during the studied time points (from day −3 until day 9 after estradiol benzoate injection). Values are presented as mean ± SEM and different superscript values indicate significant differences (*p* < 0.05).

## Data Availability

The datasets used and/or analyzed during the current study are available from the corresponding author upon reasonable request.

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
