# Peer review of "Administration of Estradiol Benzoate Enhances Ovarian and Uterine Hemodynamics in Postpartum Dairy Buffaloes"

_animals, 2023, doi:10.3390/ani13142340_

Round 1

Reviewer 1 Report

Comments on the paper:

Administration of estradiol benzoate enhances ovarian and uterine hemodynamics in postpartum dairy buffaloes. 

This paper aims to describe changes in blood flow to the uterus and ovaries, together with other haemodynamic parameters, as measured by transrectal Doppler ultrasound technique, following oestrogen injection in buffaloes approximately 30 days postpartum.   

Overall, the work is quite well thought out and well constructed. The results are comprehensively presented and the conclusions are supported by the results obtained. 

However, I found some problematic aspects that negatively affect the quality of the work as a whole. Firstly, a certain carelessness in the presentation of the purpose of the study. Why was the administration of oestrogen chosen? What are the advantages of this procedure compared to the many other, more frequently applied procedures that exist? 

Secondly, the description of the method. I suggest the authors review and improve, where possible, the description of certain measurements, e.g. the ultrasound measurement of the diameter of the uterine horn. Furthermore, no real control group was included in the project plan. Finally, and most importantly, there is a complete lack of information on the cycle status of the buffaloes involved in the study at the start of the experiment. Since the effect of estrogen differs depending on the time of the oestrus cycle, this information is extremely relevant and cannot be omitted from the description. 

I have introduced more specific comments directly in the attached PDF file. Please see the attachment. 

The work is well done and deserves consideration after appropriate editing. In particular, it is interesting and quite novel because it addresses buffaloes. I also suggest to consider expanding the discussion on the differences between bovine and buffalo. 

Some minor issues detected, but overall good quality of English language

Author Response

June 22th, 2023

Subject: Response to the Reviewers' Comments

Manuscript ID: animals-2379762

Title:  administration of estradiol benzoate enhances ovarian and uterine

hemodynamics in postpartum dairy buffaloes

Journal: Animals

Round: RI

Dear The Editor in Chief

Thank you so much for your valuable evaluation of the manuscript. The authors greatly thank the reviewers for their time and efforts during reviewing the manuscript. We hereby declare that we have addressed all comments raised by the reviewers. The modifications that were raised by the comments of the reviewers were typed in different colors (red color for Reviewer # 1 and blue color for Reviewer). The corresponding changes and refinements made in the revised paper are incorporated in the revised version of the manuscript and summarized in our response below.

-The suggestions of the referees have been incorporated in the revised manuscript submission as appended below:

Comments and Suggestions for Authors

Reviewer #1

Comments on the paper:

Administration of estradiol benzoate enhances ovarian and uterine hemodynamics in postpartum dairy buffaloes. 

This paper aims to describe changes in blood flow to the uterus and ovaries, together with other haemodynamic parameters, as measured by transrectal Doppler ultrasound technique, following oestrogen injection in buffaloes approximately 30 days postpartum.   Overall, the work is quite well thought out and well-constructed. The results are comprehensively presented and the conclusions are supported by the results obtained. 

1- However, I found some problematic aspects that negatively affect the quality of the work as a whole. Firstly, a certain carelessness in the presentation of the purpose of the study. Why was the administration of oestrogen chosen? What are the advantages of this procedure compared to the many other, more frequently applied procedures that exist?  

Reply: Thank you for your comment. We tried to enhance the presentation of the purpose of the study, and illustrate why Estrogen was chosen in the revised version of the manuscript (line 62-80 & line 93-97).

2- Secondly, the description of the method. I suggest the authors review and improve, where possible, the description of certain measurements, e.g. the ultrasound measurement of the diameter of the uterine horn. Furthermore, no real control group was included in the project plan. Finally, and most importantly, there is a complete lack of information on the cycle status of the buffaloes involved in the study at the start of the experiment. Since the effect of estrogen differs depending on the time of the oestrus cycle, this information is extremely relevant and cannot be omitted from the description. 

Reply: First, the authors greatly thank the Reviewer for this valuable comment. Secondly, we addressed our reply to this comment in the revised version of the manuscript.

For the first part of this comment, the description of all measurements in this study was improved in the revised version of the manuscript (line 125-136).

For the second part of your comment; that related to the lack of a control group in this study, the authors thought that the study was performed on a 4-week postpartum buffalo. During this stage (and up to 90 days postpartum), animals were noncyclic [4, 5] and had low blood flow to the ovary and the uterus because of the completion of uterine involution [7, 55]. Therefore, our target was to assess whether the administration of EB could alter the hemodynamic parameters of the ovary and the uterus or not. We thought addressing a control group might not add more value to assessing the studied parameters. Furthermore, the authors thought that comparing the studied parameters before and after the administration of EB is appropriate to test the effects of EB on the same individual buffalo (to avoid individual variations). However, the inclusion of a control group and extending the plan to cover the fertility potential may be important in further study and on a large-scale population of buffalo. The authors highlighted the reply to the reviewer's comment in the revised version of the manuscript (line 311-319).

For the third portion of your comment, we highlighted that all animals were acyclic at the start of this experiment (line 112-113).

References:

Krueger, L., Koerte, J., Tsousis, G., Herzog, K., Flachowsky, G., & Bollwein, H. (2009). Transrectal Doppler sonography of uterine blood flow during the first 12 weeks after parturition in healthy dairy cows. Animal reproduction science, 114(1-3), 23–31. https://doi.org/10.1016/j.anireprosci.2008.09.006

Gohar M, Zaabel S, Eldomany W, Eldosouky A, Tawfik W, Sharawy H, et al. Transrectal Doppler Ultrasound to Study the Uterine Blood Flow Changes During the Puerperium in the Egyptian Buffaloes.Journal of Advanced Veterinary Research. 2023 Jan2; 13(1),19–24.

3- I have introduced more specific comments directly in the attached PDF file. Please see the attachment. 

Reply: Thank you for your comment. We addressed your comment in the revised version of the manuscript (typed in red color).

We summarized your additional comments in the attached PDF file as appended below:

Line 15: pluriparous

Reply: Corrected (line 27)

Line 47: greater relative to what?

Reply: We addressed your comment in the revised version of the manuscript (line 56)

Line 48-50: This statement needs context clarification. Ratings of the "low, long" type are subjective and therefore need an explanation of what would be "normal." At least a clarification if the implied reference is to buffalo raised in other countries or, for example, to cattle raised in Egypt

Reply: We addressed your comment in the revised version of the manuscript (line 56-61).

Line 50: anestrus

Reply: Corrected (line 61).

Line 57: Redundant; it is clear that those are females and it is not necessary to repeat it. I suggest deleting this part

Reply: We addressed your comment in the revised version of the manuscript.

Line 59-60: I respectfully but strongly disagree. This is only valid when no corpus luteum is present. Administration of estradiol in presence of a functional corpus luteum has been shown to decrease gonadotropin secretion and induce follicular atresia, at least in dairy cattle (see for example Bo GA, Adams GP, Caccia M, Martinez M, Pierson RA, Mapletoff RJ. Ovarian follicular wave emergence after treatment with progestogen and estradiol in cattle. Anim Reprod Sci 1995; 39: 193-204.

and Bo GA, Adams GP, Pierson RA, Tribulo HE, Caccia M, Mapletoft RJ. Follicular wave dynamics after estradiol-176 treatment of heifers with or without a progestogen implant. Theriogenology 1994; 41: 1555-l 569.)

I found no indication in the method description that the presence of a functioning corpus luteum had been excluded. I suggest that this observation be considered in the introduction and/or discussion.

In general, I suggest better defining the purpose of the study: what advantage is expected from oestradiol injection over other ovulation synchronisation/induction methods?

Reply: Thank you very much for your valuable comment. We addressed your comment in the revised version of the manuscript. Refer please to Reply to comment # 2.

Line 63: The pregnancy status is not a veterinary practice. I suggest changing to "pregnancy diagnosis" or "determination of pregnancy status"

Reply: Thank you very much for your valuable comment. We addressed your comment in the revised version of the manuscript (line 82-83).

Line 63: Same as in previous comment. Development and involution are not practices. Please add "evaluation of..."

Reply: Thank you very much for your valuable comment. We addressed your comment in the revised version of the manuscript (line 82-83).

Line 76-77: Incomplete sentence: please review. Also: please explain why it is relevant to determine NO and estradiol levels.

Reply: Thank you very much for your valuable comment. We addressed your comment in the revised version of the manuscript (line 95-97).

Line 88: pluriparous

Reply: Corrected (line 108)

Line 89: What does this phrase mean? If it means that the animals were in some way already habituated to the manipulations, then this statement is irrelevant to the purpose of the experiment and should be avoided (unless a precise and standardized protocol is included in the experiment according to which the animals were habituated beforehand, but I do not believe this was the case).  

Reply: Thank you very much for your valuable comment. We addressed your comment in the revised version of the manuscript (line 109-110). We mean the following; Parturition history revealed no interventions were conducted to assist the delivery process of all animals.

Line 91: Please consider stating that all animals involved were puerperal BEFORE this point in the text, for example at the beginning of the paragraph; otherwise, reading this point it seems that there were other, non puerperal females in the experiment and it is confusing.

Reply: Thank you very much for your valuable comment. We addressed your comment in the revised version of the manuscript (line 108).

Line 92: The status of buffalo. Ovulated or not.

Reply: Animals were acyclic at the start of the experiment (line 112).

Line 93: What do you mean by vaginal redness? to my knowledge, reddening of the vaginal mucosa is also associated to oestrus, and not pathological per se

Reply: Thank you very much for your valuable comment. We addressed your comment in the revised version of the manuscript (line 114-115).

Line 94: references

Reply: Thank you very much for your valuable comment. We addressed your comment in the revised version of the manuscript (line 114-115).

Line 101-102: On this point I have a minor issue:  When did this procedure (recognizing which was the previously pregnant horn) happen? Also in the fourth week post partum, or earlier?

And how easy is it to understand which horn was pregnant in buffaloes? To my personal experience, it would not always be easy in dairy cows to tell it with certainty, if it was only assessed by rectal palpation and at any point post partum. If available, please consider inserting a reference.

Reply: The previous pregnant uterine horn had been identified earlier based on the farm records during pregnancy diagnosis using rectal palpation to differentiate the ipsilateral uterine artery from the contralateral one (Line 121-123).

Line 106: Description of the measurement

Reply: Thank you very much for your valuable comment. We addressed your comment in the revised version of the manuscript (line 125-136).

Line 163: Difference

Reply: Corrected in the whole manuscript.

Line 212: Again, preovulatory. If there is a corpus luteum the situation is different, see my comment in the introduction section

Reply: Thank you very much for your valuable comment. We addressed your comment in the revised version of the manuscript (line 242).

4- The work is well done and deserves consideration after appropriate editing. In particular, it is interesting and quite novel because it addresses buffaloes. I also suggest to consider expanding the discussion on the differences between bovine and buffalo. 

Reply: Thank you for your comment. We addressed your comment in different parts of the revised version of the manuscript (line 56-61 & line 232-233 & line 269-271 ).

5-Comments on the Quality of English Language: Some minor issues detected, but overall good quality of English language

Reply: Thank you for your comment. We tried to enhance the Quality of English Language in the revised version of the manuscript.

Once again, we thank you for the time you put into reviewing our paper and look forward to meeting your expectations. Since your inputs have been precious, in the eventuality of a publication, we would like to acknowledge your contribution explicitly.

Reviewer 2 Report

In the present work, Samir et al. try to explain that administration of estradiol benzoate enhances ovarian and uterine hemodynamics in postpartum dairy buffaloes. It is interesting, but some questions also should be explained.

1. The full stop should been deleted in the Title.

2. Editing of English language and style is needed. Pease check it throughout this manuscript.

For example, ‘The Postpartum (PP) period is a crucial factor for the resumption’ should be changed to ‘The postpartum (PP) period is a crucial stage for the resumption’.

‘(P < 0.05)’, ‘P’ should be in italic.

3. Lines 13, ‘Both E2 (pg/ml) and NO (μmol/L) significantly increased’ may be revise to ‘Both E2 and NO concentrations significantly increased’. Please check this throughout the manuscript.

4. Line 56, please explain ‘PGF2α’.

5. There are three references cited for one issue, Reference 11-13, 14-16, 17-19. Please revise them.

6. Hypothesis and aim should be added in Introduction section.

7. Lines 116, 130, ‘color maps:2 color’, ‘pulsatility indices(RI,’.

8.  Conclusion section is too simple.

9. Some references are with DOI, but Some references are not.

10. The following reference may be related to this manuscript.

Dysart LM, Messman RD, Crouse AA, Lemley CO, Larson JE. Effects of administration of exogenous estradiol benzoate on follicular, luteal, and uterine hemodynamics in beef cows. Anim Reprod Sci. 2021;232:106817.

Extensive editing of English language required.

Author Response

June 22th, 2023

Subject: Response to the Reviewers' Comments

Manuscript ID: animals-2379762

Title:  administration of estradiol benzoate enhances ovarian and uterine

hemodynamics in postpartum dairy buffaloes

Journal: Animals

Round: RI

Dear The Editor in Chief

Thank you so much for your valuable evaluation of the manuscript. The authors greatly thank the reviewers for their time and efforts during reviewing the manuscript. We hereby declare that we have addressed all comments raised by the reviewers. The modifications that were raised by the comments of the reviewers were typed in different colors (red color for Reviewer # 1 and blue color for Reviewer). The corresponding changes and refinements made in the revised paper are incorporated in the revised version of the manuscript and summarized in our response below.

-The suggestions of the referees have been incorporated in the revised manuscript submission as appended below:

Comments and Suggestions for Authors

Reviewer #2

Comments and Suggestions for Authors

In the present work, Samir et al. try to explain that the administration of estradiol benzoate enhances ovarian and uterine hemodynamics in postpartum dairy buffaloes. It is interesting, but some questions also should be explained.

  1. The full stop should been deleted in the Title.

Reply: Thank you for your comment. We addressed your comment in the revised version of the manuscript (line 3 ).

  1. Editing of English language and style is needed. Pease check it throughout this manuscript.

For example, ‘The Postpartum (PP) period is a crucial factor for the resumption’ should be changed to ‘The postpartum (PP) period is a crucial stage for the resumption’.

‘(P < 0.05)’, ‘P’ should be in italic.

Reply: Thank you for your comment. We addressed your comment in the whole revised version of the manuscript (typed in red color, for example, line 24, 42, 46, and others; type in blue color).

  1. Lines 13, ‘Both E2 (pg/ml) and NO (μmol/L) significantly increased’ may be revise to ‘Both E2 and NO concentrations significantly increased’. Please check this throughout the manuscript.

Reply: Thank you for your comment. We addressed your comment throughout the revised version of the manuscript (line 47).

  1. Line 56, please explain ‘PGF2α’.

Reply: Thank you for your comment. We addressed your comment in the revised version of the manuscript (line 236).

  1. There are three references cited for one issue, Reference 11-13, 14-16, 17-19. Please revise them.

Reply: Thank you for your comment. We addressed your comment in the revised version of the manuscript and the references were updated (Line 81-89).

  1. Hypothesis and aim should be added in Introduction section.

Reply: Thank you for your comment. We addressed your comment in the revised version of the manuscript and the introduction section was modified based on the comments raised by the two reviewers (line 62-80 & line 93-97; typed in red color).

  1. Lines 116, 130, ‘color maps:2 color’, ‘pulsatility indices (RI,’.

Reply: Thank you for your comment. We addressed your comment in the revised version of the manuscript (line 142 & 156).

  1. Conclusion section is too simple.

Reply: Thank you for your comment. We addressed your comment in the revised version of the manuscript (line 321-329).

  1. Some references are with DOI, but Some references are not.

Reply: Thank you for your comment. We addressed your comment in the revised version of the manuscript and some references were updated (line 402, 409, and 457 ).

  1. The following reference may be related to this manuscript.

Dysart LM, Messman RD, Crouse AA, Lemley CO, Larson JE. Effects of administration of exogenous estradiol benzoate on follicular, luteal, and uterine hemodynamics in beef cows. Anim Reprod Sci. 2021; 232:106817.

Reply: Thank you for your recommendation. This reference was very helpful.

Comments on the Quality of English Language: Extensive editing of English language required.

Reply: Thank you for your comment. We tried to enhance the Quality of English Language in the revised version of the manuscript.

Once again, we thank you for the time you put into reviewing our paper and look forward to meeting your expectations. Since your inputs have been precious, in the eventuality of a publication, we would like to acknowledge your contribution explicitly.

Round 2

Reviewer 1 Report

I would like to warmly thank the authors for their consideration in responding point by point to my comments. The quality of the manuscript has benefited greatly from the revision by the authors. The problems I had encountered have been completely resolved and all questions answered. 

I only added two more comments in lines 112 and 336 to the revised version. Please see the attached PDF. These two comments are intended as advice to authors for further refinement of the manuscript. No further revision is necessary and I recommend publication. 

I am flattered by the offer to explicitly publish my name in the contributions in case of publication and gratefully accept.

Author Response

Dear reviewers,

Thank you so much for your valuable evaluation of the manuscript. The authors greatly thank the reviewers for their time and efforts during reviewing the manuscript. We hereby declare that we have addressed all comments raised by the reviewers. The modifications that were raised by the comments of the reviewers were typed in red color. The corresponding changes and refinements made in the revised paper are incorporated in the revised version of the manuscript and summarized in our response below.

-The suggestions of the referees have been incorporated in the revised manuscript submission as appended below:

Comments and Suggestions for Authors

Reviewer #1

Comments on the paper:

I would like to warmly thank the authors for their consideration in responding point by point to my comments. The quality of the manuscript has benefited greatly from the revision by the authors. The problems I had encountered have been completely resolved and all questions answered.

I only added two more comments in lines 112 and 336 to the revised version. Please see the attached PDF. These two comments are intended as advice to authors for further refinement of the manuscript. No further revision is necessary and I recommend publication.

Reply: Thank you very much for your time and efforts during reviewing the manuscript. We addressed your comments in the revised version of the manuscript (typed in red color).

We summarized your comments in the attached PDF file as appended below:

Line 112: For the sake of accuracy, it would be necessary here to introduce a definition of what is meant by 'acyclic' (the term is not used unambiguously in the literature). Since it is evident from the context that the animals had not, until then, shown signs of resumption of cyclic activity, I suggest the authors briefly add this definition, perhaps in brackets, and possibly a reference. 

Reply: We addressed your comment in the revised version of the manuscript (line 112-113).

Line 336-339: I think this part still needs revising

Reply: We addressed your comment in the revised version of the manuscript (line 337-338).

Once again, we thank you for the time you put into reviewing our paper and look forward to meeting your expectations. Since your inputs have been precious, in the eventuality of a publication, we would like to acknowledge your contribution explicitly.
